# RESEE: Responding through Seeing Fine-grained Visual Knowledge in Open-domain Dialogue

**Haoqin Tu[1], Yitong Li[2,3], Fei Mi[2], Zhongliang Yang[4]***

[1]University of Chinese Academy of Sciences, [2]Huawei Noah's Ark Lab
[3] Huawei Technologies Ltd. [4]Beijing University of Post and Telecommunications
tuisaac163@gmail.com, {liyitong3, mifei2}@huawei.com, yangzl@bupt.edu.cn

## Abstract

Incorporating visual knowledge into text-only dialogue systems has become a potential direction to imitate the way humans think, imagine, and communicate. However, existing multimodal dialogue systems are either confined by the scale and quality of available datasets or the coarse concept of visual knowledge. To address these issues, we provide a new paradigm of constructing multimodal dialogues as well as two datasets extended from text-only dialogues under such paradigm (RESEE-WoW, RESEE-DD). We propose to explicitly split the visual knowledge into finer granularity ("turn-level" and "entity-level"). To further boost the accuracy and diversity of augmented visual information, we retrieve them from the Internet or a large image dataset. To demonstrate the superiority and universality of the provided visual knowledge, we propose a simple but effective framework RESEE to add visual representation into vanilla dialogue models by modality concatenations. We also conduct extensive experiments and ablations *w.r.t.* different model configurations and visual knowledge settings. Empirically, encouraging results not only demonstrate the effectiveness of introducing visual knowledge at both entity and turn level but also verify the proposed model RESEE outperforms several state-of-the-art methods on automatic and human evaluations. By leveraging text and vision knowledge, RESEE can produce informative responses with real-world visual concepts. Our code is available at https://github.com/ImKeTT/ReSee.

## 1 Introduction

With the availability of large-scale datasets (Li et al., 2017; Dinan et al., 2018) and pre-trained language models (Radford et al., 2019; Raffel et al., 2020), dialogue generation develop rapidly in recent years. Conducting effective linguistic

---
*Corresponding Author

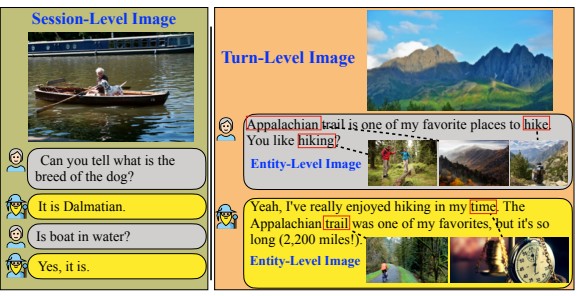

Figure 1: Traditional visual dialogue (*left*) is grounded on a single given picture, while the proposed multimodal dialogue (*right*) provides both Turn-Level and Entity-Level images based on text-only dialogue data.

communications often requires real-world experiences shared between speakers (Bisk et al., 2020). Text alone may fall short in accurately conveying rich world knowledge (Harnad, 1990), where visual signals are essential to share experiences and conduct high-quality conversations. As humans converse day to day, it is common and natural for them to group information into smaller chunks of memory through images. That explains why incorporating visual perceptions in dialogue systems can potentially bring the conversation quality to a higher level.

Visual dialogue (Das et al., 2017) was proposed to learn to communicate with users based on one simple image, making the visual knowledge very limited for a multi-turn dialogue session. In order to enhance the dialogue quality by providing larger capacity and flexibility of visual information, recent works have considered employing multiple images and image searching processes to better align with the dialogue context. Even so, they are confined to retrieving images on a coarse-grained dialogue concept (*e.g.*, session-level) or leverage inaccurate visual knowledge searched from inadequate image resources (Liang et al., 2021; Shen et al., 2021). To sum up, current works have two main issues that may compromise the performance of multimodal dialogue. (1) **Coarse-grained** visual

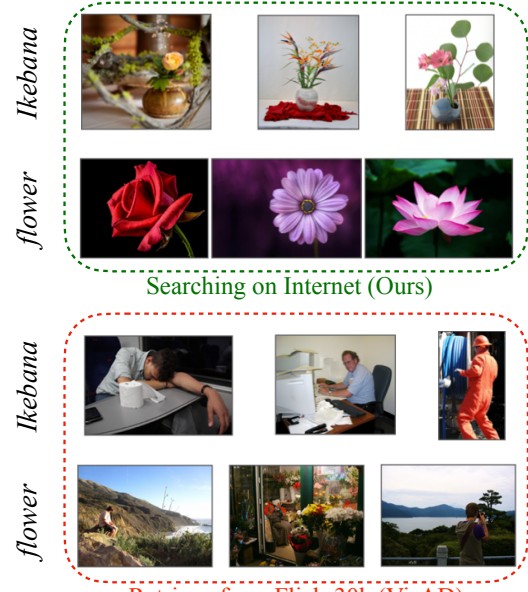

Figure 2: Samples of entity images of *Ikebana* and *flower* from searching the internet *v.s.* retrieving from limited image-caption data. Images from the internet are more **accurate** and **diverse** compared to the counterpart.

knowledge: existing multimodal dialogues mostly follow the framework of image-grounded conversation, which inherently provides insufficient visual knowledge (one image) and leaves lots of details unexploited for a complete conversation. (2) **Potentially inaccurate** visual knowledge: though recent explorations come up with using fine-grained images, they are limited in searching from small-scale image caption datasets (*e.g.*, Shen et al. (2021) employs `Flickr30k` (Young et al., 2014) for this process). These defects will introduce knowledge bias into the system (*e.g.*, entity images retrieved from `Flickr30k` may be wrong or monotonous *w.r.t.* given entities in Figure 2) and impair the conversational skills of a dialogue agent.

To overcome the above two shortcomings, we believe: (1) Compared with session-level visual knowledge, fine-grained visual knowledge such as entity-level image is more competent to help models build a comprehensive understanding of ongoing conversations. We thus propose to explicitly divide the visual standard of a dialogue session into turn-level and entity-level. (2) Instead of matching photos from existing image sets, we search images on the internet for every entity to obtain accurate and diverse visual representations accordingly. To justify the advantage of our approach in obtaining pictures with higher quality, we randomly sample 50 entities from existing dialogue data and either

search corresponding images from the internet or retrieve them from a large image corpus with over 150K images.[1] We further conduct a human evaluation to quantify entity-image relevance. Images searched from the internet outperform and tie retrieved ones in 52% and 12% cases respectively.[2] Based on the above-mentioned two concepts of visual knowledge, we take a step forward and come up with a novel framework to automatically construct multimodal dialogue data.

To verify the efficiency of provided visual information, we present RESEE, a generative conversational framework powered by real-world visual experiences. Our framework follows the encoder-decoder paradigm with either shared or separate encoder-decoder setup. We handle multimodal dialogue context by concatenating these information into the encoder, then the model generates plausible responses using its decoder. Three types of token embeddings are considered in the encoder module to sink in the knowledge from different modalities. To prove the effectiveness of RESEE, we further compare our dialogue model with several strong baselines, including four task-oriented pre-trained models and two similar multimodal dialogue systems. RESEE outperforms most baselines on both automatic and human evaluations. We also conduct comprehensive ablation experiments to demonstrate (1) the model performance gains brought by different visual knowledge, (2) the model performance with increased visual knowledge volumes, and (3) the relation between the proposed visual knowledge and the conventional document knowledge.

**Contributions.** (1) We provide a new paradigm to construct multimodal dialogue data and two datasets based on it. A comparison between ours and other multimodal dialogue datasets is in Table 1. (2) We propose a simple yet effective multimodal dialogue framework RESEE, which utilizes visual knowledge to generate informative and plausible responses. (3) Extensive experiments and promising results on two constructed datasets justify the effectiveness of our dialogue framework.

## 2 Multimodal Dialogue Datasets

In this section, we introduce our framework for constructing multimodal dialogue datasets. The overall

---

[1]We employ `COCO2017` (Lin et al., 2014) combined with `Flickr30k` as an image pool in this process.
[2]The Cohen's kappa score (Cohen, 1960) is 0.493, indicating two annotators reached a moderate agreement.

| Datasets | #Dialogues | #Utters | Domains | Auto Construct | Avg. Img. | Img. Level |
|---|---|---|---|---|---|---|
| VisDial (Das et al., 2017) | 123K | 2.4M | Image-based QAs | ✗ | 1 | Turn |
| GuessWhat (De Vries et al., 2017) | 155K | 1.6M | Image-based QAs | ✗ | 1 | Turn |
| IGC (Mostafazadeh et al., 2017) | 4K | 25K | Image-based QAs | ✗ | 1 | Turn |
| MMD (Saha et al., 2018) | 150K | 6M | Fashion Search | ✗ | 28 | Entity |
| MMConv (Liao et al., 2021) | 5.1K | 39.7K | Conv. Search | ✗ | 22.32 | Entity |
| RESEE-WoW (Ours) | 22.3K | 100.4K | Know.-based Conv. | ✔ | 24.19 | Turn&Entity |
| RESEE-DD (Ours) | 13.1K | 49.2K | Daily Conv. | ✔ | 13.83 | Turn&Entity |

Table 1: Statistic and comparison of our ReSee datasets comparing existing multimodal dialogue datasets. "Avg. Img." is the averaged number of images per dialogue session and "Img. Level" is image granularity.

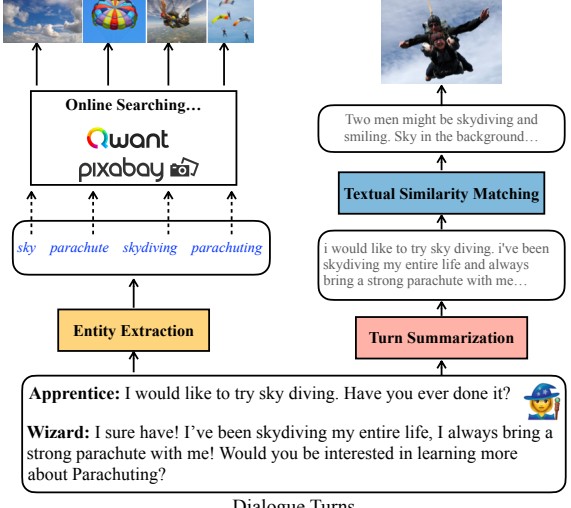

Figure 3: Data processing and construction of our dataset RESEE-WoW using one example from WoW.

data flow for dataset construction is in Figure 3. A dialogue session should consist of two aspects of visual information, namely the turn-level outline and entity-level details. We search for both visual concepts from either a very large image pool or the internet. In detail, we construct multimodal datasets extended from Wizard of Wikipedia (WoW) (Dinan et al., 2018), a knowledge-grounded dialogue dataset, and the commonly used Daily Dialogue (DD) (Li et al., 2017).

## 2.1 Turn-level Visual Knowledge

One dialogue turn is a single exchange of conversation between two speakers (*e.g.*, a question and an answer). Intuitively, turn-level visual knowledge is helpful when there are more than one topic related to a dialogue session with multiple turns, and the turn-level visual knowledge should be highly relevant to the current ongoing conversation turn.

Since one complex dialogue is generally long and diverse, instead of being restricted to one specific data domain, we gather a relatively large group of image-caption data and propose to use sentence

similarity between captions and dialogue turns for image retrieval. Using similarity from only the language domain helps us mitigate biases caused by using multimodal similarity measurement from various image domains (Liang et al., 2021).

For the image set to be searched, we group four image-caption datasets, *i.e.*, COCO2017 (Lin et al., 2014), Flickr30k (Young et al., 2014), NoCaps (Agrawal et al., 2019) and Localized Narratives (LN) (Pont-Tuset et al., 2020) with 826,539 image-caption pairs in total. Then we use the following steps for turn-level image retrieval: (1) **Turn Summarization**: To avoid information discrepancy between dialog turns and image captions arising from different sentence lengths. We first summarize the dialog turns into a shorter version. (2) **Texual Representation**: To fully leverage caption descriptions of images, we use pre-trained sentence BERT (Reimers and Gurevych, 2019) to get the textual representation of both summarized dialog turns and image captions. (3) **Image Retrieval**: Finally, we employ processed textual representations of dialogue turns as queries and representations of captions as keys to index the most relevant image to every dialogue turn from the image-caption database. And we further present the percentage of turn-level images retrieved from each image-caption dataset in Table 2.

## 2.2 Entity-level Visual Knowledge

The turn-level knowledge alone is not competent to provide full visual details for long and knowledgeable conversations. We thus propose to use entity-level images to empower the dialogue agent with insights into details. Specifically, entity-level visual knowledge involves images of both nouns and named entities from every dialogue. We use the following steps for entity extraction and their corresponding images acquirement: (1) **Named Entity**: We use a pre-trained RoBERTa model (Liu et al., 2019) to extract named entities in every dialogue

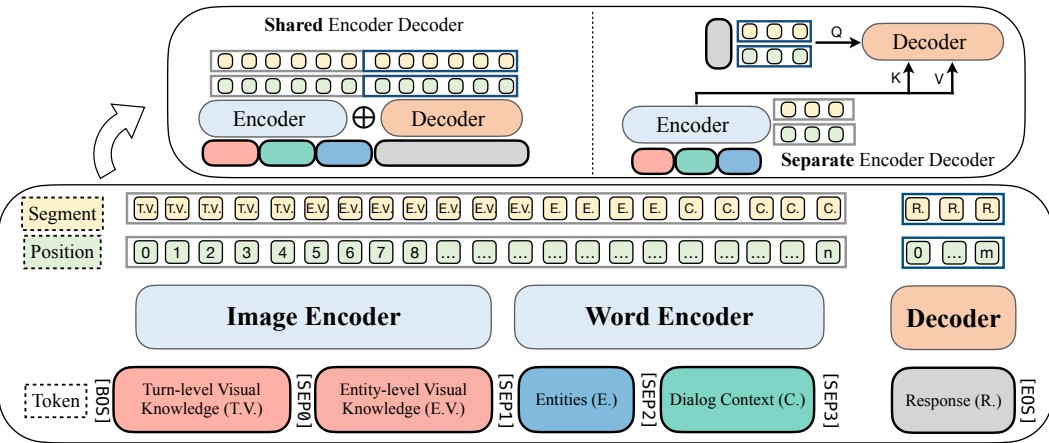

Figure 4: Model structure of the proposed multimodal dialogue system. We consider three types of embeddings (Token, Position, and Segment) as the model input. Our dialogue framework is split into two paradigms, namely model with shared encoder-decoder (RESEE (SHARE)) and model with separate encoder-decoder (RESEE (SEP.)).

| Dataset | Cap-Img | WoW (%) | DD (%) |
|---|---|---|---|
| COCO2017 (2014) | 123,287 | 2.67 | 2.99 |
| Flickr30k (2014) | 31,014 | 2.67 | 2.32 |
| NoCaps (2019) | 4,500 | 0.05 | 0.02 |
| LN2020 (2020) | 671,469 | 94.61 | 94.67 |

Table 2: Image-caption pairs in four existing datasets for turn-level image retrieval, and the percentage (%) of retrieved images from two used datasets.

instance. (2) **Regular Nouns**: We then extract all nouns from dialogues using the public toolkit Stanza (Qi et al., 2020). (3) **Image Searching**: Finally, we use two online search engines[3] to search images for the entity-level visual knowledge. Since we leverage two searching engines *i.e.*, Qwant, Pixabay in this process, we make sure that there is at least one valid image for every extracted entity.

### 2.3 Overall Dataset

The proposed datasets are advantageous in comparing prior works by providing fine-grained and more accurate images related to the dialogue context. This is because (1) we explicitly split the visual knowledge into turn-level and entity-level; (2) we use a large image pool as well as online searching engines to acquire images. We additionally present examples and detailed statistics of RESEE-WoW and RESEE-DD in Appendix B. Note that, since turn-level information is conveyed through sentences, whose semantic information may not be fully captured through conventional word matching, we did not employ online searching for turn-level images.

---

## 3 RESEE Methodology

We consider a simple approach to concatenate and to infuse multimodal information into plain dialogue models. As shown in Figure 4, we apply this approach to two transformer models with shared or separate encoder-decoder for dialogue responding.

Formally, we define our modeling task as: given the dialogue information $\{\mathbf{C}, \mathbf{E}, \mathbf{V}_T, \mathbf{V}_E\}$, where $\mathbf{C}$ is the dialogue context, $\mathbf{E}$ is the extracted entities from $\mathbf{C}$, $\mathbf{V}_T = \{V_T^1, V_T^2, .., V_T^n\}$ is a set of turn-level images from $\mathbf{C}$ and $\mathbf{V}_E = \{V_E^1, V_E^2, .., V_E^m\}$ is a set of entity-level images from $\mathbf{C}$. We aim to learn an appropriate response $\mathbf{R}$ with given information by modeling $p(\mathbf{R} \mid \mathbf{C}, \mathbf{E}, \mathbf{V}_T, \mathbf{V}_E)$.

### 3.1 Model Input

We employ different encoders for different modality encoding. In concrete, we utilize transformer blocks (Vaswani et al., 2017) for word encoding, which projects word tokens to a continuous word embedding space. For image encoding, we utilize CLIP encoder (Radford et al., 2021) to capture the global information of a picture and then use MLP functions to transform it into the same embedding space as the word. To distinguish different modality information and to identify dialogue contexts from responses, we employ three kinds of token-wise embeddings and sum them up as the input to our transformer-based dialogue systems, namely token embedding, position embedding, and segment embedding.

**Token Embedding**: The token embedding is the concatenation of $\mathbf{V}_{\mathbf{T}w}, \mathbf{V}_{\mathbf{E}w}, \mathbf{E}_w, \mathbf{C}_w, \mathbf{R}_w$, which denote the word embedding of turn-level and entity-

level visual knowledge, extracted entities, dialogue context and response respectively. We additionally add special token [SEP] between different modalities and content from distinct speakers in the dialogue. Note that, we separate response embedding $\mathbf{R}_w$ from this concatenation for the model with a separate encoder-decoder setting.

**Position Embedding**: Since the transformer model itself cannot learn the token position, we employ position embedding to encode signals of the token order in the input sequence.

**Segment Embedding**: Segment embedding is employed to differentiate which segment (turn-level or entity-level visual knowledge, textual entities, dialogue context or response) the token is in.

### 3.1.1 Model Training

**Separate Encoder-Decoder Model (RESEE (SEP.))**: Dialogue model with separate encoder decoder employs different sets of model parameters for context understanding and response generation respectively. We apply cross-attention (Vaswani et al., 2017) between the encoder output and the decoder input to bridge the gap between multimodal dialogue context learning and response generation. We first initialize it with T5 (2020) parameters. For the training objective, the model is optimized to recover the response $\mathbf{R}$ with the given multimodal knowledge $\mathbf{X} = [\mathbf{V_T}, \mathbf{V_E}, \mathbf{E}, \mathbf{C}]$:

$$\mathcal{L}_{\text{Sep}}(\mathbf{R}, \mathbf{X}) = -\sum_{w_i \in \mathbf{R}} \log p_i \left( w_i \mid \mathbf{X} \right).$$

**Shared Encoder-Decoder Model (RESEE (SHARE))**: Dialogue model with shared encoder decoder integrates the understanding and generation process with the same set of parameters. We take masked response prediction as the main training task to make the model aware of appropriate responses with multimodal dialogue context. In detail, we first initialize it with UNILM (2019). During training, 70% of the responses are replaced by a special token [MASK] or another token in the vocabulary. The masked response is denoted as $\hat{\mathbf{R}}$. In detail, we use the unmasked dialogue information $[\mathbf{X}, \mathbf{R} \backslash \hat{\mathbf{R}}]$ to predict $\hat{\mathbf{R}}$:

$$\mathcal{L}_{\text{Share}}(\hat{\mathbf{R}}, \mathbf{X}) = -\sum_{w_i \in \hat{\mathbf{R}}} \log p_i \left( w_i \mid \mathbf{X}, \mathbf{R} \backslash \hat{\mathbf{R}} \right).$$

Besides, we also follow Liang et al. (2021) to consider entity knowledge bias when decoding.

Inspired by recent progress in language generative methods (Dong et al., 2019; Wang et al., 2021), for both types of models, we process the encoder input with bi-directional attention, while giving the decoder output causal attention masks. This masking strategy makes sure our models fully understand dialogue contexts and autoregressively generate tokens with learned knowledge.

### 3.1.2 Response Generation

For the separate encoder-decoder model, we feed multimodal information $\mathbf{X}$ to the model encoder and autoregressively generate responses from the decoder. As for the shared encoder-decoder model, we first encode $\mathbf{X}$ with a special token [BOS] behind it. Then, the model starts to generate by appending a [MASK] token to the input and samples a word from the predicted distribution over vocabulary. The [MASK] token is then replaced by the generated token and a new [MASK] is appended to the input sequence for next word prediction. Both generation processes terminate when the model predicts [EOS] token or reaches the max length.

## 4 Experimental Setup

### 4.1 Evaluation Metrics

**Automatic Metrics.** We employ automatic metrics to assess the model performance:[4] (1) Fluency: perplexity (**PPL**) measures the confidence of the generated responses; (2) Token-based Relevance: **BLEU** (Papineni et al., 2002) and **Rouge-L** (Lin, 2004); Embedding-based Relevance: (Serban et al., 2017): Embedding Average cosine similarity (**Avg.**), Vector Extrema cosine similarity (**Ext.**), and Embedding Greedy Matching score (**Gre.**). (3) Diversity: Distinct-1 (**Dist-1**) and Distinct-2 (**Dist-2**) (Li et al., 2016) measure the number of distinct uni-grams and bi-grams divided by the total grams.

**Human Evaluation.** We perform human evaluation over the generated responses. We consider three conventional criteria: fluency (**Flue.**), informativeness (**Info.**), and relevance (**Relv.**) following Song et al. (2021). Also, we consider Sensibleness and Specificity Average (**SSA**) metric (Adiwardana et al., 2020), evaluating whether a response makes sense and is specific. We strictly obey a double-blind procedure, where the annotators know noth-

---

[4] All these metrics are calculated by first excluding sentence punctuation and then running the public NLG evaluation script from `github.com/Maluuba/nlg-eval`.

| Model | PPL↓ | BLEU↑ | Rouge-L↑ | Avg.↑ | Ext.↑ | Gre.↑ | Dist-1↑ | Dist-2↑ |
|---|---|---|---|---|---|---|---|---|
| | | | RESEE-WOW | | | | | |
| DIALOGPT (Zhang et al., 2020) | 13.38 | 0.0987 | 0.1467 | 0.805 | 0.423 | **0.705** | 0.163 | 0.403 |
| GPT-2 (Radford et al., 2019) | **9.64** | 0.1296 | 0.1345 | 0.793 | 0.422 | 0.677 | 0.151 | 0.371 |
| UNILM (Dong et al., 2019) | 18.86 | 0.1088 | 0.1215 | 0.770 | 0.373 | 0.582 | 0.125 | 0.335 |
| T5 (Raffel et al., 2020) | 18.35 | 0.1361 | 0.1411 | 0.830 | 0.412 | 0.620 | 0.160 | 0.392 |
| MSDP (Liu et al., 2022) | - | 0.1154 | **0.1905** | - | - | - | - | - |
| RESEE (SHARE) | 17.94 | 0.1193 | 0.1255 | 0.790 | 0.369 | 0.595 | 0.124 | 0.354 |
| RESEE (SHARE) - E. | 18.36 | 0.1183 | 0.1192 | 0.783 | 0.370 | 0.585 | 0.119 | 0.332 |
| RESEE (SHARE) - E.V. | 17.39 | 0.1167 | 0.1253 | 0.768 | 0.366 | 0.587 | 0.095 | 0.250 |
| RESEE (SHARE) - E. - T.V. | 18.54 | 0.1148 | 0.1238 | 0.778 | 0.376 | 0.593 | 0.099 | 0.269 |
| RESEE (SHARE) - E. - E.V. | 18.05 | 0.1143 | 0.1230 | 0.793 | 0.377 | 0.590 | 0.144 | 0.393 |
| RESEE (SEP.) | 17.46 | **0.1508** | 0.1599 | **0.844** | **0.426** | 0.632 | 0.162 | 0.416 |
| RESEE (SEP.) - E. | 17.47 | 0.1426 | 0.1509 | 0.837 | 0.422 | 0.627 | 0.166 | 0.430 |
| RESEE (SEP.) - E.V. | 17.58 | 0.1479 | 0.1521 | 0.841 | 0.422 | 0.629 | **0.171** | **0.440** |
| RESEE (SEP.) - E. - T.V. | 17.58 | 0.1460 | 0.1480 | 0.839 | 0.421 | 0.628 | 0.168 | 0.429 |
| RESEE (SEP.) - E. - E.V. | 17.52 | 0.1370 | 0.1456 | 0.833 | 0.420 | 0.626 | 0.161 | 0.410 |
| | | | RESEE-DD | | | | | |
| DIALOGPT (Zhang et al., 2020) | 5.95 | 0.1132 | 0.1345 | 0.734 | 0.467 | **0.658** | 0.186 | 0.466 |
| GPT-2 (Radford et al., 2019) | **5.83** | 0.1183 | 0.1519 | 0.692 | 0.434 | 0.657 | 0.134 | 0.520 |
| UNILM (Dong et al., 2019) | 6.57 | 0.0871 | 0.1031 | 0.723 | 0.424 | 0.565 | 0.115 | 0.333 |
| T5 (Raffel et al., 2020) | 8.11 | 0.1102 | 0.1392 | 0.729 | 0.469 | 0.578 | 0.183 | 0.503 |
| VISAD (Shen et al., 2021) | 17.81 | **0.1247** | - | 0.642 | 0.451 | 0.526 | 0.097 | 0.332 |
| RESEE (SHARE) | 8.94 | 0.1111 | 0.1497 | 0.773 | 0.446 | 0.607 | 0.164 | 0.446 |
| RESEE (SHARE) - E. | 9.51 | 0.0991 | 0.1444 | 0.756 | 0.451 | 0.600 | 0.168 | 0.459 |
| RESEE (SHARE) - E.V. | 10.10 | 0.0924 | 0.1301 | 0.748 | 0.434 | 0.593 | 0.126 | 0.354 |
| RESEE (SHARE) - E. - T.V. | 9.53 | 0.0873 | 0.1276 | 0.753 | 0.444 | 0.593 | 0.165 | 0.437 |
| RESEE (SHARE) - E. - E.V. | 10.02 | 0.0919 | 0.1332 | 0.749 | 0.443 | 0.594 | 0.165 | 0.447 |
| RESEE (SEP.) | 7.39 | 0.1215 | **0.1582** | **0.786** | **0.475** | 0.628 | 0.196 | 0.535 |
| RESEE (SEP.) - E. | 7.50 | 0.1204 | 0.1567 | 0.786 | 0.471 | 0.625 | 0.205 | 0.544 |
| RESEE (SEP.) - E.V. | 7.31 | 0.1197 | 0.1564 | 0.783 | 0.474 | 0.624 | **0.206** | **0.560** |
| RESEE (SEP.) - E. - T.V. | 7.66 | 0.1174 | 0.1523 | 0.767 | 0.469 | 0.618 | 0.193 | 0.528 |
| RESEE (SEP.) - E. - E.V. | 7.47 | 0.1159 | 0.1548 | 0.733 | 0.467 | 0.616 | 0.184 | 0.500 |

Table 3: Results of the proposed dialogue models on RESEE-WOW (test *unseen* set) and RESEE-DD (test set). Models without textual entity, turn-level or entity-level visual knowledge as the inputs are appended with "- E.", "- T.V." and "- E.V." respectively. Our dialogue framework RESEE with shared and separate encoder-decoder are appended with (SHARE) and (SEP.) respectively. We mark the best result with **bold face** and the second best with underline.

ing about the models. We sample 100 instances across each model for human evaluation.[5]

## 4.2 Baselines

To verify the advantages of the proposed framework in dataset construction and multimodal dialogue generation, we take competitive DIALOGPT (Zhang et al., 2020), GPT-2 (Radford et al., 2019), UNILM (Dong et al., 2019) and T5 (Raffel et al., 2020) as traditional dialogue baselines, all of which consist of 24 transformer layers. On WOW dataset, we additionally consider one recent method: MSDP (Liu et al., 2022), a dialogue model that leverages prompt tuning, multi-stage refinement with the GPT-2. On DD dataset, we incorporate a strong multimodal dialogue system VISAD (Shen et al., 2021),

which considers words extracted from dialogue context and their corresponding images into generation. Note that, RESEE (SHARE) is similar to MARIA (Liang et al., 2021), which considers similar training paradigm. However, MARIA takes only one image per dialogue session, we thus consider our RESEE (SHARE) as an extension of MARIA. See Appendix A.2, C for more model details.

## 5 Results and Analysis

**Main Results.** We present evaluation results of models with separate or shared encoder-decoder over two datasets in Table 3. (1) Our model with separate encoder-decoder (RESEE (SEP.)) performs better than the model with shared encoder-decoder (RESEE (SHARE)). This may be explained as models with separate encoder-decoder explicitly divide the understanding process of multimodal in-

---
[5]Details of human annotators are listed in Appendix E.

| Model | PPL ↓ | BLEU ↑ | Rouge-L ↑ | Avg. ↑ | Ext. ↑ | Gre. ↑ | Dist-1 ↑ | Dist-2 ↑ |
|---|---|---|---|---|---|---|---|---|
| RESEE (SEP.) w/ 1 E.V. | **7.39** | 0.1215 | 0.1582 | 0.786 | **0.475** | **0.628** | 0.196 | 0.535 |
| RESEE (SEP.) w/ 2 E.V. | 7.40 | 0.1216 | 0.1545 | 0.787 | 0.473 | 0.624 | 0.196 | 0.534 |
| RESEE (SEP.) w/ 3 E.V. | 7.62 | 0.1220 | 0.1603 | 0.787 | 0.472 | 0.624 | 0.200 | 0.536 |
| RESEE (SEP.) w/ 4 E.V. | 7.75 | **0.1246** | **0.1610** | **0.790** | **0.475** | 0.626 | 0.208 | 0.555 |
| RESEE (SEP.) w/ 5 E.V. | 7.81 | 0.1230 | 0.1502 | 0.785 | 0.469 | 0.622 | **0.211** | **0.565** |

Table 4: Model performance with varied image number per entity during training ("$n$ E.V.") over RESEE-DD.

| Model | Flue. ↑ | Info. ↑ | Relv. ↑ | SSA ↑ |
|---|---|---|---|---|
| RESEE-WoW | | | | |
| GPT-2 (2019) | 3.495 | 3.140 | 2.033 | 47.50% |
| DIALOGPT (2020) | 3.830 | **3.660** | 2.260 | 59.25% |
| UNILM (2019) | 3.745 | 3.113 | 1.800 | 41.17% |
| T5 (2020) | 3.920 | 3.495 | 2.237 | 57.50% |
| RESEE (SHARE) | 3.900 | 3.143 | 1.870 | 46.00% |
| RESEE (SEP.) | **3.930** | 3.650 | **2.267** | **64.75%** |
| RESEE-DD | | | | |
| GPT-2 (2019) | 3.916 | 3.237 | 2.760 | 50.67% |
| DIALOGPT (2020) | 4.023 | 3.547 | 2.727 | 56.17% |
| UNILM (2019) | 4.168 | 3.340 | 2.585 | 47.25% |
| T5 (2020) | 4.153 | 3.567 | 2.713 | 52.50% |
| RESEE (SHARE) | 4.187 | 3.460 | 2.587 | 48.50% |
| RESEE (SEP.) | **4.190** | **3.690** | **2.833** | **58.50%** |

Table 5: Human evaluation results.

formation and the generation of textual responses using different model parameters. This makes the model devote more to each learning phase. (2) On both constructed datasets, RESEE (SEP.) with full visual knowledge achieves the best or competitive performance in terms of relevance metrics *i.e.*, BLEU, Rouge-L, even comparing models with task-oriented pre-training (DIALOGPT) or external document knowledge (MSDP). This observation demonstrates the effectiveness of our model leveraging representations from both text and vision. (3) When considering embedding-based metrics, our method is better than baselines in **Avg.** and **Ext.**, but it is slightly inferior to two GPT models in **Gre.**. That is to say, though RESEE may not reach the similarity upper bound compared to pre-trained GPTs, it is still advantageous in the averaged sentence similarity comparing strong baselines.

We also observe that finetuned GPT-2 and DIALOGPT perform better than our method in **PPL** on both datasets. This is attributed to their pre-training stage which dedicates in directly optimizing model generation ability. However, our model can achieve better diversity compared with baselines, especially our model variants without textual entity input and/or entity-level visual knowledge.

We also present human evaluation results in Table 5,[6] which further justify the outcomes and findings from automatic metrics above.

## 5.1 Visual Knowledge

We conduct extensive ablation experiments over variants of the input information to better understand their respective roles in the dialogue generation task. (1) The performance improvement on our model benefits from both aspects of visual knowledge in providing external information. (2) Fine-grained visual information (*i.e.*, entity-level), plays a more important role in improving the generation performance than turn-level visual knowledge, which explains the necessity to find and utilize fine-grained visual clues. (3) Turn-level images also prompt model performance (*i.e.*, "-E." *v.s.* "-E.-T.V."), which is consistent with findings from the traditional visual dialogue. (4) However, textual entities bring more performance gain comparing entity-level visual knowledge. We ascribe this to the model pre-training stage that is originally on the language domain, which makes it harder for dialogue models to understand visual information than to acquire knowledge from texts. (5) Introducing visual knowledge improves the quality of generated responses, but generally degenerates the diversity. This is attributed to the constraints brought by fine-grained visual inputs. These inputs enlighten the model with explicit visual clues, making it compelling to specific knowledge but leading to a tolerable sacrifice of text diversity.

## 5.2 Multiple Entity-level Images per Entity

Since we provide a one-to-many mapping between entities in the dialogue context and their corresponding images, we conduct experiments with varied numbers of entity-level images as input. In Table 4, (1) increasing the number of entity-level images can further boost the dialogue model performance by generating more relevant responses. We ascribe this to a larger information capacity

---

[6]The average Fleiss's kappas (Fleiss and Cohen, 1973) of RESEE-WoW and RESEE-DD are 0.419 and 0.492 respectively, indicating three annotators reached a moderate agreement.

| Model | PPL ↓ | BLEU ↑ | Rouge-L ↑ | Avg. ↑ | Ext. ↑ | Gre. ↑ | Dist-1 ↑ | Dist-2 ↑ |
|---|---|---|---|---|---|---|---|---|
| T5 (Raffel et al., 2020) | 18.35 | 0.1361 | 0.1411 | 0.830 | 0.412 | 0.620 | 0.160 | 0.392 |
| T5 w/ Know. | 17.66 | 0.1411 | 0.1488 | 0.839 | 0.423 | 0.631 | 0.157 | 0.403 |
| ReSee (Sep.) | **17.46** | 0.1508 | **0.1599** | 0.844 | 0.426 | **0.632** | 0.162 | 0.416 |
| ReSee (Sep.) w/ Know. | 17.68 | **0.1528** | 0.1561 | **0.847** | **0.429** | 0.631 | **0.174** | **0.464** |
| - E. | 17.53 | 0.1463 | 0.1582 | 0.840 | 0.420 | 0.628 | 0.161 | 0.426 |
| - E. - T.V. | 17.76 | 0.1443 | 0.1485 | 0.839 | 0.424 | 0.629 | 0.168 | 0.427 |
| - E. - E.V. | 17.56 | 0.1419 | 0.1445 | 0.836 | 0.419 | 0.627 | 0.157 | 0.399 |

Table 6: Model performance with external document knowledge (" w/ Know.") on ReSee-WoW (test *unseen* set).

provided by extra visual knowledge. (2) However, giving too many entity-level images can be a show-stopper for the model, *i.e.*, the model with 5 images per entity generally performs worse. This might be attributed to the plain multimodal infusion method considered, where the model may confuse different images that belong to the same or another entity. (3) More entity-level images jeopardize the model's output confidence with lower **PPL** yet make generated responses more diverse with consistently more distinct n-grams (*i.e.*, higher **Dist-1/2**).

## 5.3 External Document Knowledge

*Is the visual knowledge a complement of existing textual knowledge?* To answer it, we conduct experiments over ReSee-WoW with provided topic passages appended to the input. In Table 6, we observe that (1) our visual knowledge can further boost model performance even with document knowledge, demonstrating the evidence provided by visual knowledge is complementary to existing textual knowledge. But the performance gain of adding documents to the visual models is not as significant as models without visual knowledge (T5). This indicates that there exist certain intersections between information provided by two modalities. (2) Bringing document knowledge to the model greatly improves diversity. Because abundant textual information helps models understand dialogues comprehensively and generate responses diversely.

## 5.4 Case Analysis

We exhibit an example of generated responses in Figure 5. As this conversation is talking about the importance of dressing code in interviews, our dataset provides one turn-level image showing a professional person with a suit and a tie as well as three entities and their corresponding images. Compared with models without visual enhancement, our two models focus more on the provided visual contexts and generate responses that are highly relevant to dialogues and the reference. For example, our

**Contexts:**
I know what you mean. I ought to wear right clothes at the right time.

You got it . Only in this way can you gain the respect of the interviewer and his confidence in your judgement.

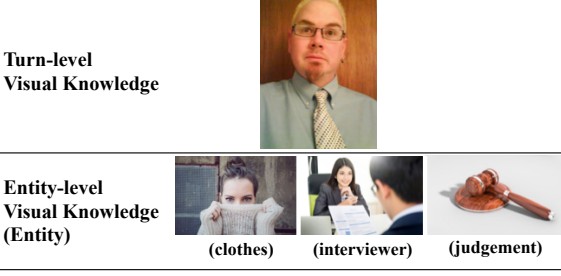

**Turn-level Visual Knowledge**

**Entity-level Visual Knowledge (Entity)**
(clothes)   (interviewer)   (judgement)

**Reference:** *It may not be true, but the first and lasting impression of you is determined by your clothes and behavior.*

**Generated Responses:**

**GPT-2** : I know that's the job.
**DialoGPT** : I think so. I think I'll wear a suit and tie.
**UniLM** : I'll try to make sure that you don't lose your interviewers.
**T5** : I agree with you.

- - - - - - - - - - - - - - - - - - - - - - - - - - - - - - -

**ReSee (Share)** : Yes, i'll try my best to make a good impression on the interviewers
**ReSee (Sep.)** : I think so. you should be comfortable in your clothes and make good impression on the interviewer.

Figure 5: An example of responses generated by our models and baselines. Highlighted words overlap entities in the dialogue context or the response reference.

models can produce words that pay more attention to "interviewer" and "clothes", which are missing in the unimodal counterparts. These demonstrate that our datasets provide useful visual information, which the proposed multimodal dialogue system captures and subsequently leverages to generate better responses that are relevant to the reference. Please refer to Appendix D for more examples.

## 6 Related Works

**Visual Dialogue Dataset.** Images can serve different purposes in a dialogue. Visual dialog (or visual question answering, VQA) is a task to answer questions about the factual contents of the image in a multi-turn manner. VisDial (Das et al., 2017) was constructed of one image and about 10 indepen-

dent question-answer pairs grounded on the given image. De Vries et al. (2017) introduced image grounded QA dataset with pixel-level object location of the image. IGC (Mostafazadeh et al., 2017) was constructed based on Twitter conversations with (image, description, question-answer) triplet as samples. In visual-enhanced conversational recommendation, MMD (Saha et al., 2018) was a multi-modal dataset under a shopping situation and aimed at providing applicable recommendations based on textual conversations as well as images of potential shopping items. MMConv (Liao et al., 2021) was applied in tourism scenarios across 5 real situations, it also provided a knowledge base and a photo gallery about recommended items. Recently, MMDialog (Feng et al., 2022) was proposed with massive multimodal open-domain conversations and associated images derived from social media. IMAD (Viktor and Denis, 2023) was constructed using massive amount of dialogues, with the last utterance to be replaced with collected images.

**Open-domain Dialogue.** Open-domain dialogue models aim at responding to general human-like conversations in various circumstances. While dialogue generation has a rich history, the area has made significant progress with the rising of pre-trained models in varied linguistic domains (Zhang et al., 2020; Mi et al., 2022; Zhu et al., 2023b; Touvron et al., 2023b). The introduction of external knowledge in traditional models plays a vital role in leading them to intellectual dialogue agents. For example, Wu et al. (2021) leveraged three domains of knowledge to enhance the model performance in Chinese contexts. Wang et al. (2022) employed an extra retrieval process to find knowledgeable evidence as input to enlarge dialogue model capacities. Recent works focus on efficient knowledge integration like retrieval-free approaches (Wang et al., 2023a) and few-shot prompting (Wang et al., 2023b). Moreover, visual knowledge has also been recently considered to boost the performance of dialogue models. Multi-Modal BLENDER (Shuster et al., 2021) was pre-trained on large-scale visual question-answer datasets for image-grounded conversation. Liang et al. (2021) introduced a method to allocate conversations with a picture as external knowledge. Shen et al. (2021) extended the visual augmentation to the token-level, providing versatile visual information to the model. Most recently, as the emergence and wide spread of large language models (LLMs), such as GPT-3 (Brown

et al., 2020), LLAMA (Touvron et al., 2023a,b), more and more works start incorporating LLMs as their text generative framework and get exceptional performance in the open-domain dialogue tasks (Zhu et al., 2023a; Liu et al., 2023; Ye et al., 2023; Dai et al., 2023).

# 7 Conclusion

In this paper, we present a paradigm for multimodal dialogue construction with two novel datasets and a multimodal dialogue responding framework RE-SEE. We explicitly separate the visual knowledge into two aspects, using online searching or retrieving from large image corpora to construct accurate and diverse visual knowledge. Transformer-based dialogue models with shared and separate encoder-decoder verify that provided visual knowledge promotes model capacity. Further, we explore feeding multiple entity-level images and external document knowledge into models. By providing fine-grained visual knowledge on dialogues, we demonstrate dialogue models can substantially achieve better performance across different setups and domains.

## Acknowledge

This work was supported in part by the National Key Research and Development Program of China under Grant 2022YFC3303301 and in part by the National Natural Science Foundation of China under Grant 6230071708 and Grant 62172053. The authors would like to thank Qiyu Wu, Haoyue Dai, and Kangwei Liu for their insightful discussions and contributing to the human evaluation process.

# 8 Limitations

(1) The provided datasets are auto-constructed, meaning visual biases brought by online searching are inevitable. We plan to take our next step to make the dataset more accurate and to include more visual knowledge (*e.g.*, visual knowledge from external document knowledge in WoW) in our multimodal dialogues. (2) For now, we did not consider a one-to-one mapping between the textual entity and entity images in the model input, more sophisticated relations can also be introduced for better modal interaction and modeling. (3) Our framework offers a novel way to enhance text-only dialogue system performance by adding extra information from a multimodal perspective. However, this comes at the cost of extra computational overhead brought by learning visual knowledge.

## 9 Ethics Statement

We are aware that automatic dialogue generation may create deceptive, harmful, or objectionable content due to their internal biases (Curry and Rieser, 2018; Gehman et al., 2020). These biases are usually inherited from the training data itself. We observe that since our dataset construction is totally based on existing text-only dialogues, our RESEE framework can be used to mitigate those biases easily. For instance, one of our future work directions is to employ the proposed multimodal data collection method on detoxification dialogues (*e.g.*, The Moral Integrity Corpus (Ziems et al., 2022)) for building safer and better dialogue agents.

We are well aware that the online searching process of entity-level images may cause biases (*e.g.*, gender, race) in our constructed dataset. To mitigate the bias, we collect multiple images on the internet for one entity in dialogues (see Appendix B for statistical details of our datasets), so that the model can choose more than one specific image during model training. For licenses of images, other employed dialogue data, and the constructed datasets that are about to be released, please refer to Appendix A.1 for more details.

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

## A  Implementation Details

### A.1  Dataset Construction

For turn-level image retrieval, we employ pre-trained BART (Lewis et al., 2020) model to summarize the dialogue turns. After we have access to representations of both dialogues and captions encoded by sentence BERT, we employ FAISS[7] for indexing speedup. As for entity-level image online searching, we use Qwant[8] and Pixabay[9] to search at least one valid image for every extracted entity. As for licences of images we employed in our datasets, Pixabay images are all royalty-free. Images from Qwant follow one of five protocols for reproduction, sharing and modification: Public domain; Non-commercial reproduction and sharing; Reproduction and sharing; Non-commercial reproduction, sharing and modification; Reproduction, sharing and modification. And our datasets will be released under Non-commercial reproduction and sharing license to ensure proper usage.

### A.2  Dialogue Models

We initialize parameters of RESEE (SEP.) and RESEE (SHARE) using T5 (Raffel et al., 2020) and UNILM (Dong et al., 2019) respectively. Note that, we only add the segment embedding to the shared encoder-decoder model to separate their respect inputs. On the RESEE-WOW dataset, we truncate the context input (*i.e.*, dialogue context, entities and visual knowledge) to a fixed length of 190, and the response to 35. We exclude the most frequent and uncommon nouns (words that appears less than 3 times and more than 100 times) to accelerate model training. The cleaned nouns in RESEE-WOW takes around $68\%$ of the original extracted words. We make sure that for every training data, the entity-level visual knowledge as well as the entity input is no more than 8 and the turn-level image is no more than 5. To make the model fully understand knowledgeable conversations in RESEE-WOW, we split every dialogue session into smaller conversational chunks with maximum of 2 turns for training. For RESEE-DD dataset, the encoder input was set to 185 with 35 to be the response. Every training data has no more than 6 entity-level images and 5 turn-level images. Also, we reduce the entity-level to around $80\%$ of the original entity-level image to accelerate training. We use AdamW opti-

---

[7] github.com/facebookresearch/faiss
[8] www.qwant.com
[9] pixabay.com

---

mizer (Loshchilov and Hutter, 2017) with the learning rate linearly increasing from 0 to 0.005 for the first $20\%$ training steps, then linearly decreasing to 0. We train the model until it has no progress on validation set (valid unseen set for RESEE-WOW). All experiments are conducted on two NVIDIA TITAN GPUs with 24G memory in total, it takes around 12 hours for RESEE-WOW training and 7 hours on RESEE-DD.

## B  Dataset Details

### B.1  Dataset Statistics

First of all, for two text-only datasets we employed, WOW dataset is under an MIT License, and it is publicly available at https://parl.ai/projects/wizard_of_wikipedia/. DD dataset is licensed under CC BY-NC-SA 4.0, and the dataset can be obtained from http://yanran.li/dailydialog. We present detailed dialogue dataset information, including unique turn-level image number, unique entity-level image amount, turn and entity level images averaged on a dialogue session and average number of images that belong to one entity in Table 7. We also show the relationship between entity number per dialogue session and dialogue session number in Figure 6, the data distribution of how many examples are there for each ($n$ entity-level image, $m$ turn-level image) setting in Figure 7. From these four distribution figures, we can tell that the RESEE-WOW dataset has more concentrated turn-level image number and entity-level image number pairs, while the range of entity-level image number of RESEE-DD is wider.

### B.2  Multimodal Examples

We present sampled examples from our constructed datasets RESEE-WOW and RESEE-DD in Figure 8. From these examples, we can clearly tell the visual enhancement for dialogue understanding from both knowing named entities and enlarging impressions of regular nouns. For instance, the noun *Ikebana* is a proper noun in the dialogue, the model would never know what it looks like from just reading the dialogue contexts. However, the entity-level image provides the model with a straightforward approach to access related visual knowledge. Another example shows that images corresponding to abstract nouns such as *love* can provide an ambiance of romance for models, which may strengthen model's

| Dataset | RESEE-WoW | | | RESEE-DD | | |
|---|---|---|---|---|---|---|
| | train | valid (*seen/unseen*) | test (*seen/unseen*) | train | valid | test |
| **Dialog Session** | 18,430 | 981/967 | 965/968 | 11,118 | 1,000 | 1,000 |
| **Turn-level Image** | 46,319 | 8,896/7,294 | 8,851/6,705 | 32,399 | 7,966 | 7,626 |
| **Entity (Image)** | 14,618 | 3,699/2,862 | 3,748/2,762 | 6,204 | 2,298 | 2,411 |
| **Avg. Turn Image** | 4.50 | 4.52/4.53 | 4.49/4.51 | 3.75 | 3.85 | 3.70 |
| **Avg. Ent. Image** | 19.87 | 18.97/18.53 | 19.00/18.66 | 9.96 | 10.16 | 10.14 |
| **Max. Ent. Image** | 60 | 44/47 | 50/48 | 67 | 76 | 46 |
| **Min. Ent. Image** | 3 | 5/4 | 5/5 | 0 | 0 | 0 |

Table 7: Statistics of two constructed multi-modal dialogue datasets. We present unique entity-level image count as well as unique image count of the 5 most similar image on turn-level visual data retrieval. The average, maximum and minimum number of images are based on one dialogue session. We also present the average number of searched valid pictures for every entity at the last row.

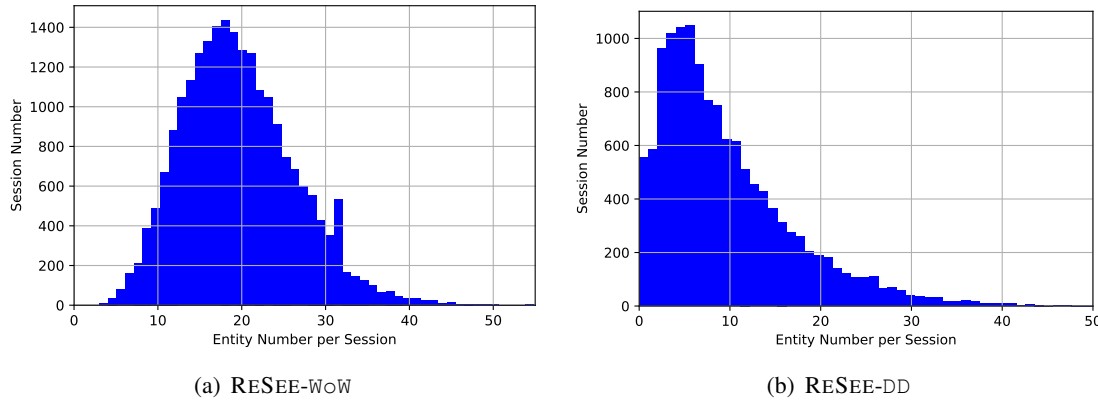

(a) RESEE-WoW

(b) RESEE-DD

Figure 6: Data distribution of entities of one dialogue session on two datasets. The X axis represents entity number, while the Y axis represents dialogue session number.

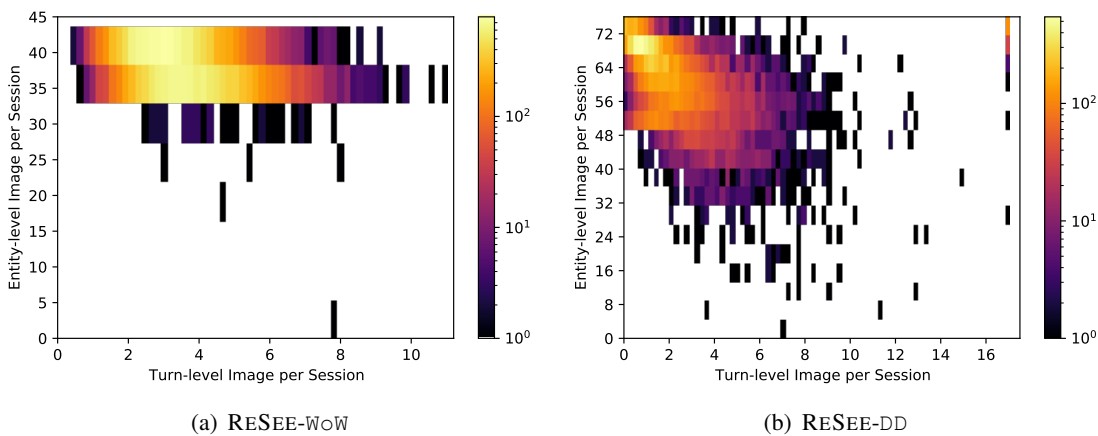

(a) RESEE-WoW

(b) RESEE-DD

Figure 7: Distribution of turn-level image and entity-level image numbers of two datasets. We use logarithm function to normalize the number of samples with varied turn-level and entity-level images and indicate their values using color bar.

understanding of dialogue histories and further assist it to produce high-quality responses.

## C Baseline Details

We present the implementation details of several baselines. We took the pre-trained weights from Huggingface for GPT-2[10] and DIALOGPT[11] model. For two models, we used their 24-layer version to make fair comparisons with rest methods.

---

[10]https://huggingface.co/gpt2-medium
[11]https://huggingface.co/microsoft/DialoGPT-medium

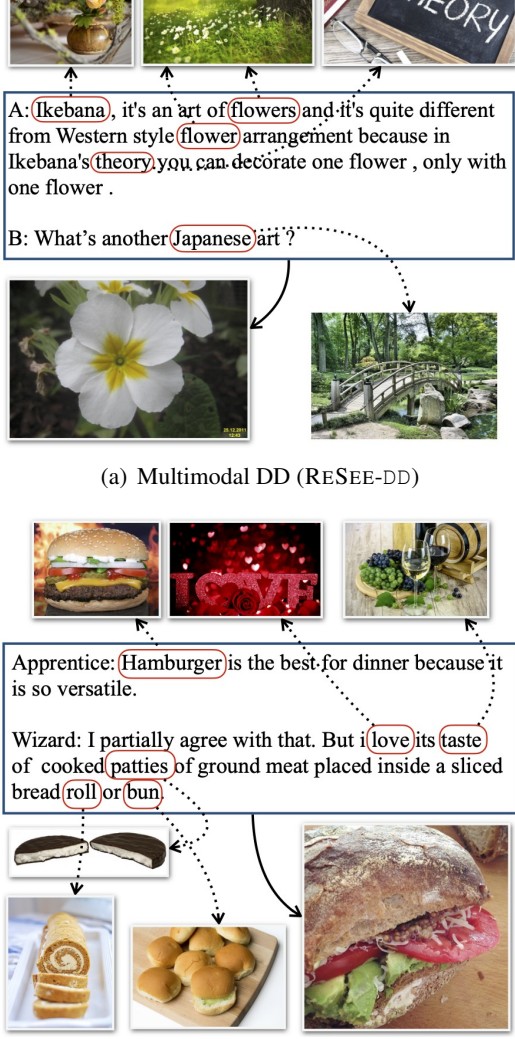

(a) Multimodal DD (RESEE-DD)

(b) Multimodal WoW (RESEE-WoW)

Figure 8: Dataset sample for one dialogue turn on our multimodal datasets. Pictures pointed by dashed lines are entity-level images, while the one pointed by solid line is turn-level image for one instance.

We used Adam (Kingma and Ba, 2014) optimizer with learning rate increases from 0 to 0.001 for the first 20% iterations for both GPT-2 and DI-ALOGPT. We truncate input data to a fixed length of 250 and make sure that the length of every generated response is no more than 30. We train two models on two datasets until they have no progress on validate sets, which takes around 3 epochs. All baselines are trained on the same machine as RE-SEE with two NVIDIA TITAN GPUs.

## D Additional Qualitative Results

We also present more generated examples of our RESEE models as well as several baseline dialogue

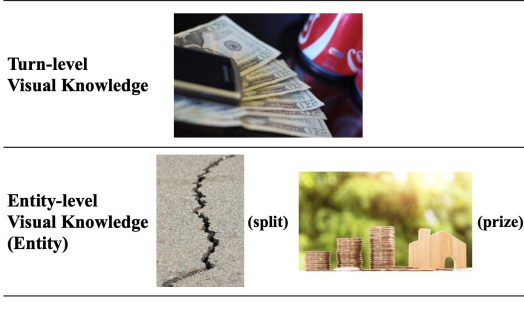

**Contexts:**
The best we can do is another 30 dollars off . That's definitely the lowest we can go . That still leaves a gap of 20 dollars .

Let's meet each other half-away again and split the difference ; I think this is a price we can both be satisfied with .

**Turn-level Visual Knowledge**

**Entity-level Visual Knowledge (Entity)**    (split)    (prize)

**Reference:** *ok . we can meet halfway again .*

**Generated Responses:**

GPT-2:    I'm sorry, but I can't do it. I'm afraid we can't make it.
DIALOGPT:    I think we can't do that.
T5:    I'm afraid we can't afford to do that anymore.
- - - - - - - - - - - - - - - - - - - - - - - - - - - - - - - - -
RESEE (SEP.): OK, let's meet again half-away and see what happens.

Figure 9: Additional example for RESEE and baselines.

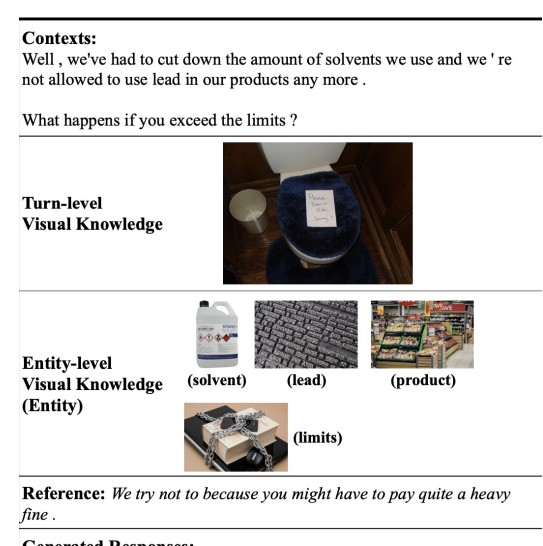

**Contexts:**
Well , we've had to cut down the amount of solvents we use and we ' re not allowed to use lead in our products any more .

What happens if you exceed the limits ?

**Turn-level Visual Knowledge**

**Entity-level Visual Knowledge (Entity)**    (solvent)    (lead)    (product)

(limits)

**Reference:** *We try not to because you might have to pay quite a heavy fine .*

**Generated Responses:**
GPT-2 :    We have to be careful about what we use.
DIALOGPT :    We have to use a lot of lead.
UNILM :    You'll be fine
T5:    I'm afraid we can't afford to do that anymore.
- - - - - - - - - - - - - - - - - - - - - - - - - - - - - - - - -
RESEE (SHARE): You'll have to pay a fine, and we will be fined $ 20 each
RESEE (SEP.):    we'll have to pay a fine if we meet the limit.

Figure 10: Additional example for RESEE and baselines.

models in Figure 9, 10, and 11. From these qualitative results, we can draw the conclusion that our RESEE method can better understand given dialogue contexts with enhanced visual knowledge, hence, generating responses with higher quality.

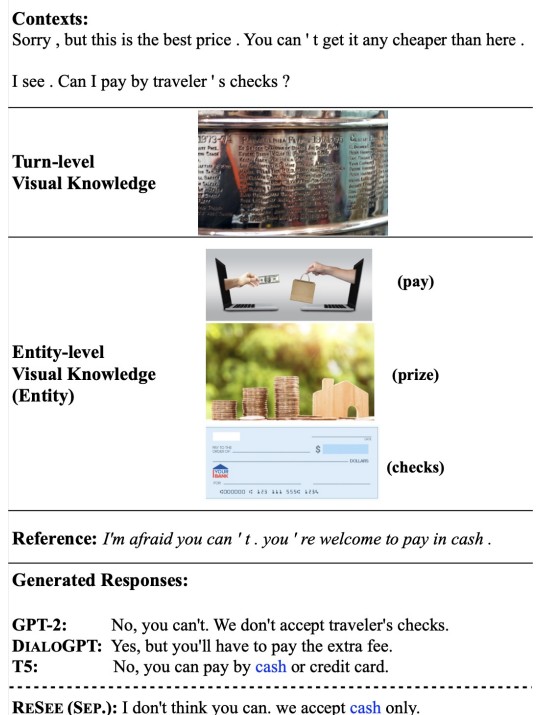

**Contexts:**
Sorry , but this is the best price . You can ' t get it any cheaper than here .

I see . Can I pay by traveler ' s checks ?

**Turn-level Visual Knowledge**

**Entity-level Visual Knowledge (Entity)**

(pay)

(prize)

(checks)

**Reference:** *I'm afraid you can ' t . you ' re welcome to pay in cash .*

**Generated Responses:**

**GPT-2:** No, you can't. We don't accept traveler's checks.
**DIALOGPT:** Yes, but you'll have to pay the extra fee.
**T5:** No, you can pay by cash or credit card.
- - - - - - - - - - - - - - - - - - - - - - - - - - - - - - - - - - - - - - - - -
**RESEE (SEP.):** I don't think you can. we accept cash only.

Figure 11: Additional example for RESEE and baselines.

# E    Human Evaluation

For annotators, we hire three undergraduate students from America or China with fluent English reading skills. Each annotator is assigned 100 (instances)×6 (models)×4 (aspects) = 2,400 rating tasks, resulting in 2,400 (tasks)×3 (annotators) = 7,200 human ratings in total. The annotators have acknowledged the use of annotated data sets and are paid an average annotation salary. All annotators were aware of the potential risks or ethical concerns of machine-generated texts.

**Annotation Instruction**    Here we present the human evaluation standard:

**Fluency:**

1. The system's result does not make sense and it is unreadable.

2. Choose this score when you are hesitant between score 1 and score 3.

3. The system's result contains minor errors but they do not affect your understanding.

4. Choose this score when you are hesitant between score 3 and score 5.

5. The system's result is human-like, grammatically correct, and very easy to understand.

**Informativeness:**

1. The system's result is dull, repetitive, and does not have new information.

2. Choose this score when you are hesitant between score 1 and score 3.

3. The system's result contains some new information and it displays a certain level of diversity.

4. Choose this score when you are hesitant between score 3 and score 5.

5. The system's result is very informative and contains novel content. In addition, it displays a high level of diversity and it is enjoyable to read.

**Relevance:**

1. The system's result is completely irrelevant to the given reference.

2. Choose this score when you are hesitant between score 1 and score 3.

3. The system's result is partially related to the reference and some of its content can be found in the reference.

4. Choose this score when you are hesitant between score 3 and score 5.

5. The system's result is very related to the given reference and contains a diverse set of concepts in the reference.

**Make Sense:**

- **YES**: the response is completely reasonable in context.

- **NO**: the response is confusing, illogical, out of context, or factually wrong.

**Being Specific**

- **YES**: the response is specific to the given context.

- **NO**: the response could be used in dozens of different contexts.