# OpenReview forum: "ReSee: Responding through Seeing Fine-grained Visual Knowledge in Open-domain Dialogue"
_EMNLP/2023/Conference — EMNLP 2023 Main_

### Official Review · Reviewer_ks6W · 2023-07-29

**Soundness:** 4

**Excitement:**

4: Strong: This paper deepens the understanding of some phenomenon or lowers the barriers to an existing research direction.

**Paper Topic And Main Contributions:**

The authors present a novel approach that enhances text-only dialogue systems by integrating visual knowledge, addressing limitations in the existing solutions. These limitations include the restricted scale and quality of available datasets, as well as less aware of the visual knowledge integration. To overcome these challenges, the authors introduce a new paradigm for constructing multimodal dialogues. Their method involves a deliberate and explicit division of visual knowledge into finer granularities at both the entity and turn levels. These finer-grained visual representations are obtained from either the Internet or a large image dataset. The proposed framework, named RESEE, seamlessly incorporates visual representations into vanilla dialogue models. Despite its straightforward nature, RESEE proves to be highly effective in enhancing dialogue systems. Extensive experiments and ablations are conducted, demonstrating the efficacy of integrating visual knowledge and confirming the superiority of the RESEE model over alternative approaches. Additionally, the authors contribute to the field by providing two datasets that extend text-only dialogues. These datasets hold promise for further advancements in this area of study.

**Questions For The Authors:**

1. At the beginning of this paper, I think Figure 1 could use a bit more explanation. It's essential because the figure really helps demonstrate the novelty and significance of the authors' work.

2. I'm curious about how well this approach would work with less informative prompts.

3. I am also curious about this approach's efficiency. Though the work shows a competitive quantitative performance, it is also important to benchmark this work's computational cost.

4. This work offers a strong performance but I think to reimplement it might concurs a few challenges due to its complexity. Would authors be making this work along with the dataset open-accessible in future?

**Reasons To Accept:**

The authors' work represents a significant and exciting contribution to the fields. The article is well-structured and written, demonstrating a clear motivation, detailed methodology explanations, and a well-designed experimental setup.

The study focuses on addressing a prevalent limitation in contemporary text-only dialogue systems. These limitations encompass restricted scale and quality of available datasets, as well as insufficient integration of visual knowledge. The authors propose a novel paradigm for constructing a multimodal dialogue system that effectively incorporates visual knowledge into various granularities.

Furthermore, the authors make a valuable contribution to the field by introducing two datasets that extend the scope of text-only dialogues. These datasets hold promise for advancing research in this domain.

This work offers extensive ablation studies, examining different variations of input information to gain deeper insights into their respective roles in the dialogue generation task. Additionally, the study presents human evaluation results, further reinforcing and justifying the findings obtained from automatic metrics. Such human evaluation yields a potential for real-application, such as multi-modal based E-commerce recommender system.

**Reasons To Reject:**

Authors should enrich the paper with qualitative results that offer a comprehensive and illustrative assessment of the model's performance, enabling a deeper understanding of its capabilities, limitations, and areas for enhancement.

Further, the authors have not addressed the cold-start performance of their work. Given the likelihood of real-world applications encountering user prompts beyond the model's learned response domain distribution, investigating cold-start scenarios becomes imperative.

Additionally, a bit concern of its work's reproducibility as this is a fairly complex approach. Thus, incorporating a clear and concise algorithmic box, illustrating the learning algorithm employed for training and inference, would be beneficial in providing readers with a more transparent view of the methodology.

**Reproducibility:**

3: Could reproduce the results with some difficulty. The settings of parameters are underspecified or subjectively determined; the training/evaluation data are not widely available.

**Reviewer Confidence:**

4: Quite sure. I tried to check the important points carefully. It's unlikely, though conceivable, that I missed something that should affect my ratings.

---

> ### Author Rebuttal · Authors · 2023-08-29
>
> Thank you for your time and feedback. For your concerns:
>
> - Qualitative results: thank you for your advice, due to page limits, we were unable to present more qualitative examples in the main part; we will append more results in the appendix to show the limitations and other capacities of our models in our final version.
> - Cold-start performance: thank you for bringing up the important discussion of cold-start performance. However, this work may not directly aim to address this issue; we acknowledge its importance and plan to consider it as a future step in our research. We believe that investigating cold-start scenarios will enhance the practical applicability of our work, and we look forward to exploring this area in future studies.
> - Reproducibility: we have provided the training/inference code in the Supplementary Materials. We will release the proposed datasets and codes, and we also plan to maintain and improve the datasets by eliminating potential biases in the data.
>
> For your questions:
> - About Figure 1: thanks for your advice, we will add more explanations to Figure 1.
> - Less informative prompts: in our work, the prompt we feed into the dialogue system is the dialogue context, entity phrases, and two levels of images. In Table 3, we present comprehensive ablation results for different information inputs. With less informative prompts (*i.e.*, without images or entity texts), the dialogue models perform worse on both datasets. Moreover, we investigate the model performance gain brought by an increased number of entity-level images per entity in the dialogue in Table 4, which are considered to represent detail information. And there exists an upper bound of adding such information, *i.e.*, more than 4 entity-level images inversely impairs model capability.
> - Efficiency: we have listed the total training time of each of our models in Appendix A.2, since our models are initialized using UniLM and T5 severally, our dialogue models can be trained at a relatively small cost (no more than 12 hours using two GPUs with 48G memory in total).
> - Accessibility: we will release the datasets, codes, and model checkpoints for future research.

---

### Official Review · Reviewer_oqrU · 2023-08-01

**Soundness:** 3

**Excitement:**

3: Ambivalent: It has merits (e.g., it reports state-of-the-art results, the idea is nice), but there are key weaknesses (e.g., it describes incremental work), and it can significantly benefit from another round of revision. However, I won't object to accepting it if my co-reviewers champion it.

**Missing References:**

Line369: Maria; It would be beneficial to cite the first introduction of a model or approach, even if it has been previously cited in a different context.

**Paper Topic And Main Contributions:**

This paper proposes a new approach to collect images for multimodal dialogues. Specifically, two types of images (turn-level and entity-level) are collected for higher fidelity. Simple, but effective baseline model is also presented. Extensive experiments demonstrate the effectiveness of the new datasets and model.

**Questions For The Authors:**

What is the purpose of extending the dialogue datasets with images? i.e., What is the role of the images in the dialogues?

**Reasons To Accept:**

- The paper proposes better ways to collect high-quality images for multimodal conversation
- The papers collect finer-grained images for improving accuracy.

**Reasons To Reject:**

- The novelty of the proposed approach seems weak. As also mentioned in the paper, it appears to be merely a simple extension of previous work. Moreover, this extension itself does not seem particularly interesting or innovative. The reliance on larger datasets and online image search might not constitute a significant contribution.

- Although the paper, as claimed, proposes an effective model, the performance improvement doesn't seem significant.

**Reproducibility:**

4: Could mostly reproduce the results, but there may be some variation because of sample variance or minor variations in their interpretation of the protocol or method.

**Reviewer Confidence:**

4: Quite sure. I tried to check the important points carefully. It's unlikely, though conceivable, that I missed something that should affect my ratings.

---

> ### Author Rebuttal · Authors · 2023-08-29
>
> Thank you for your time and feedback. For your concerns:
>
> - Weak novelty: our main contributions are centered on two sides: 1. We propose a new paradigm for constructing multimodal dialogue data and **two new multimodal datasets**; 2. a framework to incorporate multimodal inputs and generate dialogue responses with **two new multimodal dialogue models**.  Our dataset construction framework is unique by considering both fine-grained entity-level and turn-level images as knowledge augmentation to text-only dialogues. On the model side, only our ReSee (share) is regarded as an extension of Maria with finer-grained visual input (L370-372). Note that (1) our proposed ReSee (Sep.) stands out from other existing dialogue methods due to its use of encoder-decoder architecture with designed multimodal input and training. (2) Both of our models are specifically designed to handle visual knowledge on entity- and turn-level in the dialogue, which further verify the efficacy of the proposed visual knowledge even with a simple modality integration method.
>
> - Significant contribution of larger datasets and online searching: First of all, the benefit of larger multimodal datasets is to improve model training with larger scale. Secondly, online image searching is one initial step of our data construction paradigm (L93-101), while building upon online searching, we have further developed a comprehensive end-to-end multimodal data construction framework to ensure the data quality and more fine-grained image-dialogue data. Through this framework, we demonstrate the effectiveness of our approaches of dataset crafting and two dialogue models.
>
> - Insignificant improvement in model: Table 3 illustrates the superiority of our proposed models with visual enhancement over vanilla UniLM and T5 models (the dialogue model upon which our ReSee models are initialized) by up to 33% and 18% on ReSee-DD dataset using the Rouge-L metric, respectively.  This significant improvement highlights the effectiveness of incorporating the proposed visual knowledge, even through simple integration methods, which has surpassed some SOTA dialogue models (*e.g.*, MSDP on BLEU, VisAD on embedding-based metrics).
>
> For your questions:
>
> - The purpose of extending the dialogue with images: the primary purpose of extending the text-only dialogue data with images is to enhance model performance by incorporating external knowledge. As we are motivated in the Introduction section: Visual knowledge can accurately convey rich world knowledge beyond text, providing commonsense knowledge or related information that may not be acquired by specific nouns or entities through text alone (L38-44). In practical situations, it may be difficult to classify a named entity solely based on words at the first glance. For example, models may not understand what an "ikebana" is by just looking at the word, but an image of "ikebana" can help text-only dialogue systems quickly link this concept with flowers and arts (Figure 2), allowing these models to make more informed associations and better understand underlying meaning of novel concepts. Additionally, in Section 5.3, we demonstrate that our provided visual knowledge can still enhance dialogue systems even with rich textual knowledge.
>
> - The role of images: the fundamental role of images in our work is a knowledge augmentation of existing dialogue contexts. In particular, turn-level and entity-level images serve to provide the comprehensive information (*e.g.*, the skydiving event in Figure 3) of a conversation and depict details or associated scene in the chat (*e.g.*, the sky, the parachutes), respectively.
>
> In addition, thanks for pointing out the missing references, we will check and improve them.

---

### Official Review · Reviewer_cBSK · 2023-08-04

**Paper Topic And Main Contributions:** 1) Providing a new paradigm of constr…
**Typos Grammar Style And Presentation Improvements:** N/A
**Soundness:** 3

**Excitement:**

4: Strong: This paper deepens the understanding of some phenomenon or lowers the barriers to an existing research direction.

**Missing References:**

N/A

**Questions For The Authors:**

N/A

**Reasons To Accept:**

Two new dataset RESEE-WoW and RESEE-DD
A simple but effective method RESEE to add visual representation into vanilla dialogue models by modality concatenations


**Reasons To Reject:**

The datasets are auto-constructed. Maybe there are visual biases.


**Reproducibility:**

4: Could mostly reproduce the results, but there may be some variation because of sample variance or minor variations in their interpretation of the protocol or method.

**Reviewer Confidence:**

4: Quite sure. I tried to check the important points carefully. It's unlikely, though conceivable, that I missed something that should affect my ratings.

---

> ### Author Rebuttal · Authors · 2023-08-29
>
> Thank you for your time and feedback. For your concerns:
>
> - Visual biases: thank you for pointing out this limitation. We recognize that dataset biases are a critical issue and we do not take them lightly. In fact, we explicitly acknowledge the presence of certain noises in the auto-constructed datasets and intend to take steps to make them more accurate and inclusive of visual knowledge, such as external document knowledge (L513-519) as the future work.

---

### Meta-Review · Area_Chair_cTaf · 2023-09-15

**Recommendation:** 4

**Metareview:**

Paper proposes a "simple but effective" method to add visual knowledge into vanilla dialogue data. Two new visually augmented datasets are provided as an outcome of the method.

**Pros**: Most reviewers agree the contributions of the work, and in particular, the datasets provided by the work will be a good resource to the community. One reviewer comments explicitly on the experimental study, citing components (e.g., ablation and human evaluation) that provided confidence in takeaways and potential real-world application. Multiple reviewers find the work to be fairly exciting/interesting.

**Cons**: One reviewer has concerns about the presence of visual biases - given that the dataset is automatically constructed. Another provides examples of experiments that would improve the work. Authors make some promises to ease concerns, but leave some challenges as future work. Reviewers appear to generally consider these technical concerns as minor after rebuttal.

With regards to novelty, there is some disagreement. While two reviewers find the work to be exciting/interesting - one of which takes care to highlight many of the strengths of the work - another reviewer finds the approach to be more incremental, emphasizing that the reliance on larger data sources may not be a significant contribution. Given the disagreement, I looked into the rebuttals and some paper details as well. From a methodological standpoint, I agree "larger data source" is not a significant contribution. *But,* some novelty is also owed to the use of refined visual knowledge (entity + turn level). Indeed, this refinement in approach leads to two new datasets which are well positioned as novel by the authors (Table 1).

---

### Decision · Program_Chairs · 2023-10-07

**Decision:**

Accept-Main

**Comment:**

Paper proposes a "simple but effective" method to add visual knowledge into vanilla dialogue data. Two new visually augmented datasets are provided as an outcome of the method.

**Pros**: Most reviewers agree the contributions of the work, and in particular, the datasets provided by the work will be a good resource to the community. One reviewer comments explicitly on the experimental study, citing components (e.g., ablation and human evaluation) that provided confidence in takeaways and potential real-world application. Multiple reviewers find the work to be fairly exciting/interesting.

**Cons**: One reviewer has concerns about the presence of visual biases - given that the dataset is automatically constructed. Another provides examples of experiments that would improve the work. Authors make some promises to ease concerns, but leave some challenges as future work. Reviewers appear to generally consider these technical concerns as minor after rebuttal.

With regards to novelty, there is some disagreement. While two reviewers find the work to be exciting/interesting - one of which takes care to highlight many of the strengths of the work - another reviewer finds the approach to be more incremental, emphasizing that the reliance on larger data sources may not be a significant contribution. Given the disagreement, I looked into the rebuttals and some paper details as well. From a methodological standpoint, I agree "larger data source" is not a significant contribution. *But,* some novelty is also owed to the use of refined visual knowledge (entity + turn level). Indeed, this refinement in approach leads to two new datasets which are well positioned as novel by the authors (Table 1).